# Prospective Study Comparing Outcome following Complete Polypropylene Suture Ligation versus Partial Thin Film Band Attenuation of Congenital Portosystemic Shunts in Dogs

**DOI:** 10.3390/vetsci10070480

**Published:** 2023-07-23

**Authors:** Victoria Lipscomb, Chloe Cassie, Ben Ritchie, Stephen Greenhalgh, Mickey Tivers

**Affiliations:** 1Department of Clinical Sciences and Services, Royal Veterinary College, University of London, Hatfield AL9 7TA, UK; ccassie7@rvc.ac.uk (C.C.);; 2Paragon Veterinary Referrals, Wakefield WF1 2DF, UK; mickey.tivers@paragonreferrals.co.uk

**Keywords:** congenital, portosystemic shunt, liver, surgery, portovenography

## Abstract

**Simple Summary:**

A congenital portosystemic shunt is an abnormal vessel that bypasses the liver. Dogs born with this shunt may display various clinical signs due to substances that are usually metabolized by the liver being present in much higher levels in the bloodstream. Surgery to narrow or close the shunt is recommended. Materials placed to narrow the shunt aim to close the shunt fully over time by creating an inflammatory reaction around the shunt, although some dogs may also receive a second surgery if this is not successful. Multiple acquired shunts may also develop following the narrowing or closure of a congenital shunt; these represent the opening of previously non-functional vessels to relieve increased pressure within the liver. The main objective of this study was to report the outcome for dogs treated with a ‘complete ligation where possible’ philosophy. The second aim was to compare the outcome between complete polypropylene suture ligation and partial thin film band narrowing of the congenital extrahepatic portosystemic shunt in dogs. Dogs that could not tolerate acute complete shunt ligation at surgery received partial shunt narrowing with a thin film band. Of the 110 dogs enrolled, 57 received complete ligation and 53 received partial thin film band narrowing of the shunt. Peri-operative mortality, the occurrence of post-attenuation neurological complications, the occurrence of multiple acquired shunts, the postoperative clinical shunt score and quality of life score was not significantly different between the two groups.

**Abstract:**

The main objective was to conduct a prospective study reporting the outcome for dogs with an extrahepatic congenital portosystemic shunt (CPSS) treated with a ‘complete ligation where possible’ philosophy. The second aim was to compare the outcomes following complete (C) polypropylene suture ligation versus partial thin film band (TFB) attenuation of a CPSS in dogs. Dogs that could not tolerate acute complete shunt ligation at surgery received partial shunt attenuation with TFB. Peri-operative complications, mortality, follow-up imaging findings, pre- and post-operative bile acid stimulation test results and details of any revision surgery performed were recorded. A follow-up health-related quality of life questionnaire enabled the calculation of a postoperative clinical shunt score, a quality of life score, and determined if any dogs were still on a hepatic diet and/or other medical management at a minimum of 6 months after surgery. Of the 110 dogs enrolled, 57 received complete ligation and 53 received partial TFB attenuation. Peri-operative mortality, the occurrence of post-attenuation neurological complications, the occurrence of multiple acquired shunts, the postoperative clinical shunt score and quality of life score were not significantly different between the two groups. Dogs in the C group were older, heavier and demonstrated a greater number of shunt classifications where the entry into the systemic circulation was the phrenic vein or azygous vein. Dogs in the TFB group had a greater number of unchanged bile acid concentrations after surgery, were more likely to remain on the hepatic diet and/or medical management after surgery and underwent a greater number of revision surgeries. There was variability in the precision of both ultrasound and computed tomographic angiography follow-up imaging compared to intra-operative mesenteric portovenography findings at revision surgery. Overall, dogs with an extrahepatic portosystemic shunt receiving either complete acute shunt ligation or partial TFB shunt attenuation are expected to have an excellent long-term clinical outcome and there is no reason to suggest that a dog able to tolerate complete acute shunt closure should be denied the benefit of this.

## 1. Introduction

Surgical treatment of a congenital portosystemic shunt (CPSS) aims to restore portal hepatic perfusion by CPSS closure [1]. Placement of a thin film band (TFB), with or without initial narrowing of the shunt, aims to induce the gradual and eventual complete occlusion of the vessel [2,3,4,5,6,7,8,9,10]. One study concluded that shunts should be narrowed to 3 mm or less to induce complete CPSS closure using TFB [2]. Another study that compared dogs with CPSS narrowed to < 3 mm using TFB to dogs that had TFB applied without any initial shunt narrowing reported no difference in post-operative clinical grades between the two groups and concluded that initial shunt narrowing with TFB was not required [4]. Another study narrowed CPSS in dogs using TFB to a minimum diameter that could be safely tolerated based on intra-operative parameters that assessed portal hypertension (including portal pressure and pancreas/intestinal cyanosis) and achieved 75% complete shunt closure based on follow-up intra-operative mesenteric portovenography [7]. Other studies that placed TFB around CPSS in dogs without any initial narrowing reported between 17% to 71% persistent shunting on follow-up imaging, with most dogs being clinically improved [5,6,8,9,10]. An issue when trying to compare rates of shunt closure between centers is the use of different sources of non-proprietary thin film band which introduces variability and may contribute to the wide range of reported persistent shunting following TFB placement around a CPSS. The optimal method and material for TFB CPSS attenuation remains unclear and the clinical significance of a small degree of postoperative residual shunt flow in dogs that have been treated surgically for a CPSS is unknown. Dogs with TFB placed around their CPSS may be more likely to need revision surgery than dogs with an ameroid constrictor placed around their CPSS, although both groups are reported to achieve favorable long-term outcomes [8,9].

Multiple acquired shunts (MAS) have been reported in up to 20% of dogs following surgical treatment of their CPSS [3,6,11], and there is minimal information to suggest whether the method or degree of shunt attenuation is associated with the development of MAS. Post-attenuation neurological signs (PANS) have been reported to occur in up to 12% of dogs undergoing surgical treatment of a CPSS [11,12,13,14,15,16,17]. Method or degree of extrahepatic shunt attenuation has not been shown to be a risk factor for PANS [18].

Many studies reporting outcomes following surgical treatment of a CPSS in dogs do not categorize or exclude dogs that can tolerate complete acute ligation of their shunt. Given that complete shunt closure is universally associated with an excellent long-term outcome it appears reasonable to attempt complete acute CPSS ligation in dogs that can safely tolerate this and reserve gradual attenuation methods for the remaining dogs [1,7,19,20,21,22,23,24]. A study comparing peri-operative outcomes between dogs receiving ameroid constrictor placement, and partial or complete silk ligation of their CPSS reported no significant difference in peri-operative mortality between the groups; the group allocation was based on surgeon preference, ameroid constructor availability and not the dog’s ability to tolerate complete acute CPSS ligation [25]. In a study where the selection criteria only included dogs that were not able to tolerate complete acute shunt ligation, there was no peri-operative mortality in either of the groups, which received partial attenuation of their shunt with either TFB or polypropylene [7]. To our knowledge, there are no studies directly comparing outcomes between a population of dogs with an extrahepatic CPSS that can tolerate complete acute shunt ligation and those that can only tolerate partial TFB attenuation.

The main aim of this study was to report the outcome of dogs with an extrahepatic CPSS treated with a ‘complete ligation where possible’ philosophy. The second aim was to directly compare the outcome between the dogs able to tolerate complete acute ligation of their (C) shunt to dogs only able to tolerate partial thin film band attenuation of their shunt (TFB). Our hypotheses were (1) long-term (> 6 months) postoperative clinical shunt score and quality of life score will not differ between C and TFB, (2) peri-operative mortality and occurrence of PANS will not differ between C and TFB, (3) occurrence of MAS will not differ between C and TFB, (4) TFB will have a greater number of revision surgeries, (5) TFB will have a greater number of dogs postoperatively on a hepatic diet and/or medical management of their shunt, (6) Postoperative pre and post-prandial bile acid concentrations will be significantly higher for TFB than C.

## 2. Materials and Methods

This prospective study included all dogs receiving surgical treatment of a congenital extrahepatic portosystemic shunt between January 2017 and Sept 2022. Owners provided written consent for their dog’s involvement in the study and ethical approval was granted by the participating institution (URN SR2018-1656).

The information recorded included age at surgery, breed, sex, neuter status, bodyweight and pre-operative bile acid stimulation test (BAST) results. Shunts were classified into four categories based on their entry into the systemic circulation: (1) gastro/spleno-caval shunts, (2) gastro/spleno-phrenic shunts, (3) gastro/spleno-azygos shunts, (4) colo-caval/iliac shunts [26]. Peri-operative (admission to discharge) mortality and complications, including occurrence and grade of PANS, was recorded. Dogs were deemed to have PANS if any neurological signs were noted after surgery and before discharge. PANS were graded as mild, moderate, or severe using a previously published grading scheme [16].

Dogs underwent CPSS attenuation according to a standardized surgical protocol [22]. Treatment was not randomized but was allocated based on intra-operative clinical assessment The shunt was temporarily completely occluded during surgery and changes that could be attributed to portal hypertension were assessed. Alterations in mean arterial pressure, central venous pressure, heart rate, portal pressure ≥ 14 mmHg, ETCO_2_ ≥ 20% from starting values, as well as intestinal or pancreas cyanosis, were considered evidence of portal hypertension. Dogs without any of these significant alterations during temporary full shunt occlusion intra-operatively had their shunts fully ligated with polypropylene suture (Ethicon; Johnson & Johnson) (C group). Dogs in which evidence of temporary portal hypertension developed received narrowing of their shunt with an autoclaved triple folded four mm wide, clear, non-proprietary, synthetic polymer thin film band with a uniform thickness (mean 55.3 μm, SD 18.33) (TFB group). Intra-operative mesenteric portovenography (IOMP) confirmed the correct positioning of the complete ligature or TFB around the shunt vessel close to its insertion on the vena cava or azygous vein, cranial to any contributing shunt branches. TFB was secured around the shunt using at least four metal clips. The metal clips were positioned to narrow the shunt as far as possible without inducing any of the alterations described above suggestive of portal hypertension. A loose polypropylene suture was left in situ around the shunt in case it might be needed for a future surgery. A polypropylene suture was chosen for shunt ligation because it is non-absorbable; previous use of silk at the author’s institution resulted in one dog where a silk suture placed around a shunt was re-absorbed leading to the need for revision surgery due to the recurrence of shunting.

A follow-up pre and postprandial bile acid concentration was performed approximately three months after surgery. For the purpose of describing all of the factors that owners used to decide whether or not to pursue follow-up imaging and/or a revision surgery, follow-up bile acids were categorized into three groups: (1) Normal: both pre and postprandial bile acid concentrations within the reference interval, (2) Improved: bile acid concentrations markedly reduced compared to before surgery, at least one value not within the reference interval, no value >100 μmol/L, (3) Unchanged: bile acid concentrations very high, similar to before surgery, at least one value > 100 μmol/L.

Follow-up ultrasound and/or computed tomography angiography (CTA) approximately three months after surgery was recommended for all dogs with unchanged or reduced bile acid concentrations, especially if accompanied by persistent clinical signs. The time from surgery to imaging was recorded and imaging findings after surgery documented the presence or absence of residual flow through the original shunt and the presence or absence of MAS.

Owners were asked to fill out a Health Related Quality of Life (HRQoL) questionnaire at least six months after surgery, from which a quality of life (QoL) score and a clinical congenital portosystemic shunt (CPSS) score were calculated. Scores for each clinical sign (except drinking/urination) were allocated as follows: never = 0; less than once a month = 1; monthly = 2; weekly = 3; daily = 4. Scores for drinking and urination were allocated as follows: normal = 0; unsure = 1; slightly more than usual = 2; definitely excessive = 3; severely excessive = 4. The CPSS clinical scoring method used was a modification of one previously described [27] where clinical signs were weighted according to their impact on quality of life. The clinical signs included and their multiplication factors for CPSS score calculation are presented in Table 1. The modifications to our HRQoL questionnaire from the previous study [27] were the additional inclusion of a question on frequency of hypersalivation (scored as a class 2 clinical sign) and two questions relating to drinking and urination, respectively, (scored as class 3 clinical signs). Our CPSS clinical scoring method resulted in a maximum score of 126. The QoL score was a single question within this questionnaire in which the owner was asked to rate their pets’ quality of life both before and after surgery from 1 (worst imaginable) to 10 (best imaginable). The HRQoL questionnaire is detailed in the Appendix A.

Revision surgery was recommended for dogs with unchanged bile acids and persistent flow through the original shunt on follow-up imaging, especially if accompanied by persistent clinical signs. Revision surgery was considered, at the discretion of the owner, for dogs with reduced bile acid concentrations and persistent flow through the original shunt on follow-up imaging, especially if accompanied by persistent clinical signs. Further surgery was contra-indicated in dogs with MAS. The time from the first to second surgery was recorded and findings at second surgery documented the presence or absence of residual flow through the original shunt and/or the presence or absence of MAS using intra-operative mesenteric portovenography (IOMP) and whether the shunt could be completely ligated at the second surgery.

Statistical analysis was performed using a statistical software program (IBM SPSS Statistics Version 28). Prior to the study, we performed a sample size calculation to determine the number of dogs required to find a statistically significant difference in QoL score and clinical shunt score between the two treatment groups. We estimated that 75% of the shunts treated with TFB would close completely based on data that was subsequently published [7]; using this assumption, at least 60 cases (30 per group) were required, assuming a beta of 0.8 and an alpha of 0.05. Categorical data were reported as percentages and compared with Chi square or Fisher’s exact tests. Continuous data was assessed for normality using the Shapiro-Wilk Test. Median and range were reported for non-parametric data and this was analyzed using a Mann–Whitney U test. Differences were considered statistically significant at *p* < 0.05.

## 3. Results

One hundred and ten dogs were prospectively enrolled in the study, 57 received complete suture ligation (C) and 53 received partial TFB attenuation (TFB). Intra-operative mesenteric portovenography (IOMP) confirmed optimal placement of C or TFB around a single congenital extrahepatic shunt in 109 dogs. One dog had two entry points of their shunt into the caudal vena around the level of the epiploic foramen and received partial TFB of the most cranial branch at the first surgery followed by complete polypropylene suture ligation of both branches at a second surgery six months later.

There were 29 dogs (51%) with gastro/spleno-caval shunts, 17 dogs (30%) with gastro/spleno-phrenic shunts, 11 dogs (19%) with gastro/spleno-azygous shunts and no dogs with colo-caval/iliac shunts in the C group. There were 44 dogs (83%) with gastro/spleno-caval shunts, two dogs (4%) with gastro/spleno-phrenic shunts, seven dogs (13%) with gastro/spleno-azygous shunts and no dogs with colo-caval/iliac shunts in the TFB group. The number of dogs with shunts inserting into either the phrenic or azygous vein, as opposed to the vena cava was significantly greater in the C group (28/57; 49%) compared to the TFB group (9/53; 17%) (*p* < 0.001).

There were 31 males in the C group, 21 of whom were neutered. There were 26 females in the C group, 19 of whom were neutered. There were 26 males in the TFB group, 12 of whom were neutered. There were 27 females in the TFB group, six of whom were neutered. Breeds in the C group included nine ShihTzus, eight crossbreeds, seven Miniature Schnauzers, seven Pugs, four Dachsunds, three Border Terriers, two Toy Poodles, two Jack Russell Terriers, two Chihuahuas and one each of Cavapoo, Cavachon, Pomeranian, Lhaso Apso, Bichon Frise, Cocker Spaniel, Maltipoo, Cairn Terrier, King Charles Cavalier Spaniel and Papillon. Breeds in the TFB group included seven Pugs, seven Yorkshire Terriers, five crossbreeds, four Jack Russell Terriers, four West Highland White Terriers, three Miniature Schnauzers, three Chihuahuas, two Shih Tzus, two Bichon Frises, two Cavachons and one each of King Charles Cavalier Spaniel, Papillon, Dachsund, Irish Setter, Italian Greyhound, Havenese, Cocker Spaniel, Labradoodle, Kerry Blue Terrier, Nova Scotia Duck-Tolling Retriever, Russian Tsvetnaya Bolonka, Lurcher, Jackapoo and Cockerpoo. The median age at first surgery was 32 months (range 4 months to 123 months) in the C group. The median age at first surgery was 11 months (range 2 months to 74 months) in the TFB group. Age at the first surgery in the C group was significantly greater than the TFB group (*p* < 0.001). Median bodyweight at first surgery was 6.2 kg (range 1.4 kg to 24 kg) in the C group. Median bodyweight at first surgery was 5.2 kg (range 1.6 kg to 23.1 kg) in the TFB group. Bodyweight at first surgery in the C group was significantly greater than the TFB group (*p* = 0.020).

PANS occurred in six dogs (10.5%, five grade 3) in the C group compared to seven dogs (13.2%, two grade 3) in the TFB group at first surgery. The occurrence of PANS was not significantly different between groups (*p* = 0.771). There were no PANS after a second surgery in any dog.

Two dogs died (3.5%) peri-operatively in the C group compared to two dogs (3.8%) in the TFB group at the first surgery. Peri-operatively mortality at the first surgery was not significantly different between groups (*p* = 0.941). The cause of death for one dog in the C group and two dogs in the TFB groups was grade 3 PANS that did not respond to aggressive medical treatment. One dog in the C group was arrested and died of acute portal hypertension due to a suspected portal thrombus seen on postoperative ultrasound the day after surgery. One dog in the TFB group was arrested and died the evening following a second surgery due to an unstable anaesthesia and postoperative recovery that included hypotension, hypovolaemia, melaena, anuric acute kidney injury, hypoglycaemia, epistaxis, regurgitation and respiratory acidosis. This dog had multiple acquired shunts (MAS) identified at the second surgery. The overall surgical mortality was two dogs (3.5%) in the C group compared to three dogs (5.7%) in the TFB group. Overall surgical mortality was not significantly different between groups (*p* = 0.671).

Postoperative pre and postprandial serum bile acid concentrations were performed for 34/55 dogs (61.8%) in the C group and 39/51 dogs (76.5%) in the TFB group at a median of 3.5 (range 1 to 29) months and 3.5 (range 1 to 47) months after surgery, respectively (Figure 1). There was no significant difference in the completion of pre- and postprandial bile acid concentration (*p* = 0.142) or time interval to completion of these tests (*p* = 0.955) between the two groups after surgery. For dogs that had more than one surgery the results analyzed were those after the first surgery prior to a second surgery. Median postoperative preprandial bile acid concentration was 2.9 μmol/L (range 0.0 to 190.0) for C and 10.7 μmol/L (range 0.2 to 198.6) for TFB, this difference was not statistically significant (*p* = 0.082). Median postoperative postprandial bile acid concentration was 23.0 μmol/L (range 3.3 to 476.7) for C and 36.0 μmol/L (range 0.2 to 316.2) for TFB, this difference was not statistically significant (*p* = 0.271). Using the categorization outlined in the methods, nine dogs (26.5%) in the C group and 10 dogs (25.6%) in the TFB group had normal follow-up bile acid concentrations (Figure 1). Twenty four dogs (70.6%) in the C group and 18 dogs (46.2%) in the TFB group had improved bile acid concentrations (Figure 1). One dog (2.9%) in the C group and 11 dogs (28.2%) in the TFB had unchanged bile acid concentrations (Figure 1). There was a statistically significant difference between the two groups (*p* = 0.009).

Postoperative HRQoL questionnaires were completed for 40/55 dogs (72.7%) dogs in the C group and 36/51 dogs (70.8%) in the TFB group at a median of 15 (range 6 to 49) months and 17 (range 6 to 71) months after surgery, respectively. For dogs that had a second surgery, the questionnaires were completed after the first surgery prior to undergoing the second surgery. There was no significant difference in completion (*p* = 0.832) or time interval to completion of the HRQoL questionnaire (*p* = 0.34) after surgery between the two groups. The median postoperative QoL score was 10 (range 5 to 10) for C and 10 (range 5 to 10) for TFB, this difference was not statistically different (*p* = 0.844). Median postoperative clinical shunt score was 6 (range 0 to 52) for C and 4.5 (range 0 to 32) for TFB, this difference was not statistically different (*p* = 0.260), with seven dogs in the C group and nine dogs in the TFB displaying one or more of class 1 (seizures) or class 2 (other neurological signs) clinical signs long-term.

Four dogs out of 40 dogs (10.0%) were still on a hepatic diet and/or medical management in the C group compared to 16/36 dogs (44.4%) in the TFB group, this difference was statistically significant (*p* < 0.001). In the C group, two dogs on a hepatic diet and/or medication had long-term neurological signs and one dog on a hepatic diet had unchanged bile acid concentrations. In the TFB group, seven dogs on a hepatic diet and/or medication had clinical signs, five dogs had improved bile acid concentrations and three dogs had unchanged bile acid concentrations. The reason(s) that one dog in the C group and six dogs in the TFB group remained on hepatic diet and/or medical treatment without clinical signs and with normal or improved bile acid concentrations is unknown.

Four out of 55 dogs (7.3%) in the C group returned for follow-up imaging at a median of 12.5 months (6 to 37 months) after surgery (Figure 2). Three dogs in the C group (two ultrasound and one CTA) had no residual flow or MAS identified (Figure 2). One dog in the C group had residual shunting (no MAS) identified on CTA and underwent a second surgery where no residual shunting or MAS was found on IOMP (Figure 2).

Twenty-five dogs out of 51 (49.0%) in the TFB group returned for follow-up imaging at a median of 4 months (3 to 10 months) after surgery (Figure 3). Ten dogs (40%) in the TFB group had no residual flow in their shunt or MAS identified (five ultrasound, four CTA and one that received both ultrasound and CTA) (Figure 3). Fifteen dogs (60.0%) in the TFB group had persistent shunting on imaging (seven ultrasound, six CTA and two that received both ultrasound and CTA) (Figure 3). All TFB dogs with persistent shunting had flow through their original shunt (Figure 3). MAS were not seen on follow-up ultrasound or CTA imaging in any of the TFB dogs, although MAS were documented later in one dog on IOMP at a second surgery (this dog had CTA follow imaging) (Figure 3). Of the TFB dogs that had persistent shunting on follow-up ultrasound or CTA imaging, this was confirmed with IOMP at the second surgery in seven dogs (three ultrasound, three CTA and one that received both ultrasound and CTA) (Figure 3). Three dogs (one ultrasound, one CTA and one that received both ultrasound and CTA) in the TFB group with residual flow through their shunt identified on imaging (no MAS) underwent a second surgery where no residual shunting was found on IOMP (Figure 3). In five dogs (three ultrasound and two CTA) in the TFB group with residual flow through their shunt identified on imaging, the postoperative bile acid concentrations were improved and the dogs were clinically improved so the owners opted not to pursue a second surgery.

One dog (1.8%) out of 55 in the C group underwent a revision surgery compared with 12 dogs (23.5%) in the TFB group, this difference was statistically significant (*p* < 0.001) (Figure 4). The second surgery was performed at a median of 6 months (range 6 to 13) after the first surgery. Eleven of the 13 dogs that underwent revision surgery, including the one dog in the C group, had residual flow through their original shunt identified on follow-up imaging prior to the second surgery (Figure 4). Two of the 13 dogs that had revision surgery did not have any follow-up imaging prior to the second surgery; one dog was known to have two shunts, only one of which had been attenuated with TFB at the first surgery and so a second surgery was planned, the other dog was operated on again without follow-up imaging for financial reasons.

Of the 11 dogs with persistent shunt flow identified prior to the second surgery, six had unchanged bile acid concentrations, including the one dog in the C group, and five had improved bile acid concentrations. One dog in the TFB group with residual flow and unchanged bile acids had neurological signs that remained unchanged compared to before surgery and the remaining dogs were clinically improved compared to before surgery.

Intra-operative mesenteric portovenography (IOMP) was performed in all 13 dogs at the second surgery (Figure 2, Figure 3 and Figure 4). Of the 11 dogs with residual shunt flow identified on imaging prior to the second surgery IOMP demonstrated that four dogs (including the one C group dog) had progressed to complete closure of their shunt contrary to the findings of follow-up ultrasound or CTA imaging, whilst in the other seven dogs their shunts were still patent (Figure 2, Figure 3 and Figure 4). Of these seven dogs, five had their shunt completely ligated with the loose polypropylene suture that had been left in situ at the first surgery (Figure 4). One dog that had received partial TFB only tolerated further partial shunt occlusion with suture (Figure 4), this was the dog with both unchanged neurological signs and unchanged bile acid concentrations after the first surgery. One dog had persistent shunting through the original shunt and MAS identified on IOMP, no further shunt attenuation was performed (Figure 4), and this dog died postoperatively.

Two dogs had a second surgery and IOMP without pre-operative imaging (Figure 3). One dog had persistent shunting through the original shunt and a second shunt on IOMP, both shunts were completely ligated with polypropylene suture (Figure 4). The other dog had MAS and a closed original shunt on IOMP and no further shunt attenuation was performed (Figure 4).

Overall, MAS were identified in two TFB dogs (3.9%) and no C dogs (Figure 2 and Figure 3). The difference in the occurrence of MAS between the two groups was not significantly different (*p* = 0.329). The rate of MAS for the entire study population was 2/110 dogs (1.8%).

## 4. Discussion

This prospective study reports a combined approach to treatment, with dogs divided into two groups receiving different treatments, complete ligation or partial thin film band attenuation of their CPSS, based on intra-operative assessment of portal hypertension during temporary complete shunt occlusion. This is the first study to directly compare outcomes between dogs that could tolerate complete acute ligation of their extrahepatic CPSS (‘complete ligation if possible philosophy’) to dogs that could only tolerate partial thin film band attenuation of their shunt, including initial acute narrowing of the shunt with TFB to a minimum diameter based on intra-operative assessment of portal hypertension.

Peri-operative mortality of 3.5–3.8%, occurrence of PANS 10.5–13.2% and postoperative development of MAS 0–3.9% reported in this study compared favourably with previous publications of dogs undergoing surgery for an extrahepatic portosystemic shunt using a variety of techniques [1,3,4,5,6,7,8,9,10,11,12,13,14,15,16,17,18,28] and did not differ between C and TFB groups. The lack of difference between groups in our study is interesting given that dogs that can tolerate complete acute ligation are likely to have well-developed intrahepatic portal vasculature (indicating a lower risk for intrahepatic portal pressure to be exceeded) and a better ability to tolerate perioperative drugs and anaesthetics compared to dogs that cannot tolerate complete acute ligation. The intra-operative parameters used to assess the development of portal hypertension appear to have a wide safety margin as the occurrence of portal hypertension-related complications following either complete shunt ligation or partial TFB shunt attenuation with initial acute shunt narrowing in this study was low: one dog in the C group died of a suspected portal vein thrombus seen on ultrasound the day after surgery and no dogs in either group had revision surgery to loosen a ligature or thin-film band.

Dogs in the C group were older and heavier which seems intuitive if one assumes that dogs presenting with clinical signs later in life are likely to do so because they have better developed normal intrahepatic vasculature alongside their shunt and can therefore often tolerate complete acute shunt ligation. The dog’s age or body weight at the time of surgery has not been correlated with post-operative mortality or long-term outcome after CPSS attenuation [11,29,30]. In one study, for every increase in 1 kg of body weight, the odds of developing a post-operative complication decreased by 30% [9]. Interestingly, dogs in the C group also demonstrated a greater number of shunt classifications where the entry into the systemic circulation was the phrenic vein or azygous vein. This is similar to a study in which dogs with splenophrenic and splenoazygos shunts tended to be diagnosed at later ages than dogs with right gastric-caval and spleno-caval shunts [31], as well as another study which concluded that splenophrenic and splenoazygos shunts were less clinical than dogs with shunts inserting on the vena cava caudal to the liver [32]. This supports the hypothesis that these shunt types may be partially compressed by the diaphragm during respiration and by gastric distension after eating leading to improved hepatic perfusion via the portal vein [32].

There was no difference in postoperative clinical shunt score or quality of life score between C and TFB at a median follow-up time of 15 months and 17 months, respectively, although dogs in the TFB group were more likely to remain on the hepatic diet and/or medical management after surgery. This suggests that in the majority of dogs with residual shunt flow following TFB placement, the residual shunt flow is not clinically significant within the follow-up time periods achieved in this study. However, some dogs remained on medical management and this may have contributed to the reason(s) for the lack of difference in clinical outcome between the two groups. It is also possible that owner subconscious bias due to substantial improvement of their pet from a poor baseline quality of life prior to surgery and/or emotional/financial investment in their pet contributed to a higher quality of life scores, although this bias would presumably affect both C and TFB groups. Furthermore, one study reported that recurrence of clinical signs relating to residual shunt flow (following partial suture ligation of the shunt) occurred at a mean follow-up of 36.2 months after surgery [33] therefore our study would have ideally needed to achieve double the length of follow-up period to confirm the equal long-term clinical success between C and TFB beyond 18 months.

Whilst the number of normal and improved bile acid concentrations after surgery were similar between the C and TFB groups, dogs in the TFB group had a greater number of unchanged bile acid concentrations after surgery. A previous study demonstrated that whilst bile acid concentrations do not often return to within the reference intervals after complete ligation of an extrahepatic shunt, the majority are significantly improved at a mean of 12 months after surgery and these did not subsequently change significantly over longer-term follow-up at a mean of 158 months [34]. Unchanged bile acids, therefore, represent a concern for persistent shunting, either through the original shunt or due to the development of MAS, and along with residual flow through the original shunt identified on follow-up imaging, contributed to the decision making that resulted in a greater number of TFB undergoing revision surgery compared to those in the C group. Although the broad categorization of postoperative bile acid concentrations into ‘normal’, ‘improved’ or ‘unchanged’ is of course limited by large variability within the ‘improved’ category in particular, it was helpful for owners assimilating large amounts of complex information, including variable follow-up imaging findings, when making decisions about whether or not to pursue revision surgery.

In the current study, there was a very high rate of persistent shunting seen on follow-up imaging in the TFB group at 60% when compared to previous studies [5,6,8,9,10]. The reason for this difference could be the study design, in our study dogs with less well-developed intrahepatic portal vasculature are likely to have been included in the TFB group. Our study also demonstrated a relatively low rate of MAS (3.9% in the TFB group, 0% in the C group; 1.8% overall) compared to previous studies, which is interesting given that in our study the TFB was used to acutely narrow the shunt as far as possible rather than being placed loosely. It is possible that variability in the composition of different sources of TFB contributed to this observation.

Unfortunately, our study also demonstrated variability in the precision of both ultrasound and computed tomographic angiography (CTA) follow-up imaging to identify both shunt patency and MAS compared to intraoperative mesenteric portovenography findings (IOMP) findings at revision surgery. Post-operative imaging to identify residual flow through the original shunt or MAS has been highlighted as critical to thoroughly assess outcomes and differentiate between surgical techniques for CPSS treatment in dogs [23]. The ideal timing and method of follow-up imaging after shunt surgery is yet to be defined and future prospective studies should focus on comparing larger numbers of dogs undergoing post-operative ultrasound, CTA and splenoportography.

The findings of this study highlight a lack of consistency between the various outcome measures used. Discrepancies between clinical outcomes, bile acid concentrations and different imaging modalities mean that the best method for assessing outcomes is unclear. The authors believe that the goal of surgery is to abolish shunting but the amount of residual shunting that can be considered ‘acceptable’ for a good outcome is yet to be elucidated.

Only one of the 13 dogs that underwent revision surgery had unchanged neurological signs of hepatic encephalopathy and it may be that other dogs (with no or improved clinical signs) operated on for persistent flow through their shunt were operated on prematurely in this study and would not have developed recurrence of serious clinical signs in the long-term. Investigation of the clinical significance of residual shunt flow should also be the focus of future research, utilizing clinical shunt score and quality of life score as tools for assessing long-term success when the decision is made not to re-operate on dogs with documented residual shunt flow on imaging.

Limitations of this study include the fact that multiple surgeons and anaesthetists participated, albeit following the prospective study design and standard patient management protocol. The sample size was estimated to be adequate to assess the main hypotheses based on power calculations prior to commencing the study. However, greater group sizes may have allowed more detailed conclusions to be made. Despite the study being prospective and all owners signing consent for their involvement, not all owners filled out the HRQoL questionnaire, and filling out the questionnaire was not blinded. Additionally, not all dogs returned for a follow-up bile acid stimulation test and/or repeat imaging; in particular, it is not possible to justify follow-up imaging in all dogs that are clinically well, especially if they have received a complete shunt ligation. Whilst median long-term follow-up times of 15 and 17 months were achieved in this study, longer follow-up times (over three years) would have been desirable and should be the focus of future studies.

## 5. Conclusions

This study provides the first direct comparison of outcomes for dogs receiving complete shunt ligation versus partial TFB shunt attenuation and provides detailed information that is useful for owners considering surgical treatment of their dog’s extrahepatic shunt. Dogs with an extrahepatic CPSS receiving either complete acute shunt ligation or partial TFB shunt attenuation are expected to have an excellent long-term clinical outcome. However, there is no reason to suggest that a dog able to tolerate complete acute shunt closure should be denied the benefit of this, especially as these dogs are less likely to remain on medical management, less likely to have unchanged bile acid concentrations post-operatively and less likely to undergo revision surgery.

## Figures and Tables

**Figure 1 vetsci-10-00480-f001:**
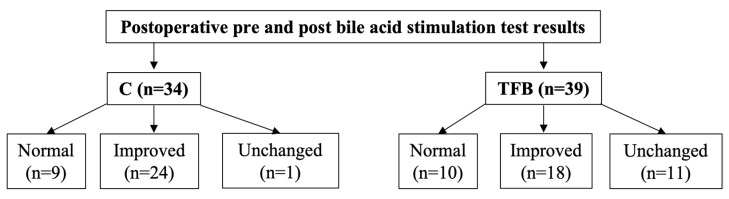
Postoperative pre and post-prandial bile acid stimulation test results by category for dogs in C and TFB groups. Normal: both pre and postprandial bile acid concentrations within the reference interval. Improved: bile acid concentrations markedly reduced compared to before surgery, at least one value not within reference range, no value >100 μmol/L. Unchanged: bile acid concentrations very high, similar to before surgery, at least one value > 100 μmol/L.

**Figure 2 vetsci-10-00480-f002:**
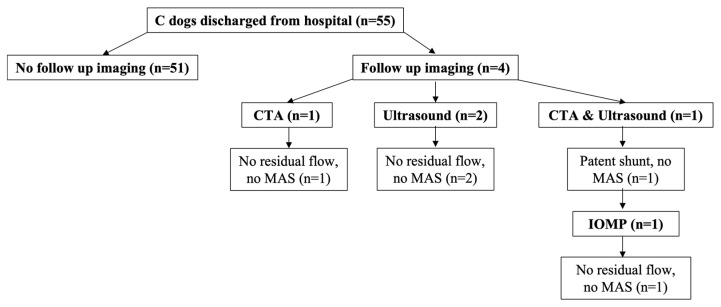
Follow-up imaging findings after first surgery and intra-operative mesenteric portovenography (IOMP) findings at second surgery for dogs in the C group.

**Figure 3 vetsci-10-00480-f003:**
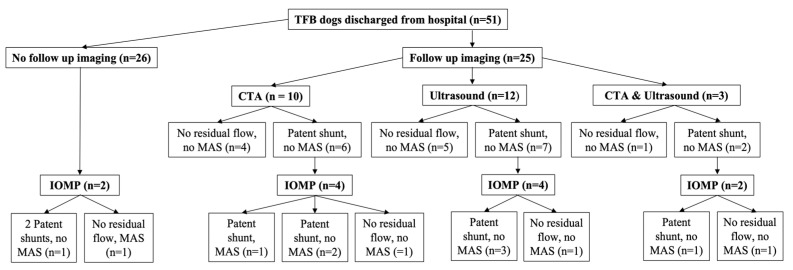
Follow-up imaging findings after first surgery and intra-operative mesenteric portovenography (IOMP) findings at second surgery for dogs in the TFB group.

**Figure 4 vetsci-10-00480-f004:**
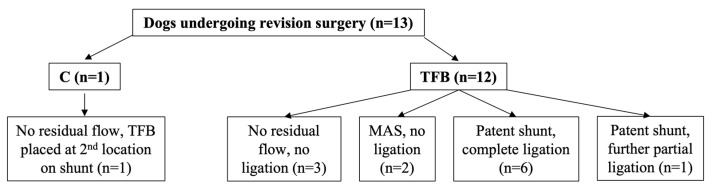
Intra-operative findings at revision surgery for dogs in C and TFB groups.

**Table 1 vetsci-10-00480-t001:** Clinical signs categorized according to impact on quality of life with different weightings for calculation of congenital portosystemic shunt (CPSS) score.

Class of Clinical Sign	Clinical Sign	Score Weighting
1	Seizures	3 × CPSS score
2	Head-pressing, circling, disorientation, aggression, collapse, unresponsiveness, apparent blindness, hypersalivation, lethargy/weakness	2 × CPSS score
3	Vomiting, diarrhoea, inappetence, polyuria, polydipsia, haematuria, dysuria	1 × CPSS score
N/A	Urolithiasis or urethral obstruction	+2 to CPSS score if present
N/A	Retarded growth	+2 to CPSS score if unsure +4 to CPSS score if present

N/A: did not fit into a class of clinical sign.

## Data Availability

The data presented in this study are available on request from the corresponding author. The data are not publicly available due to this being a small non-funded study.

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
