# Peer review of "Prospective Study Comparing Outcome following Complete Polypropylene Suture Ligation versus Partial Thin Film Band Attenuation of Congenital Portosystemic Shunts in Dogs"

_vetsci, 2023, doi:10.3390/vetsci10070480_

Round 1
Reviewer 1 Report
Dear authors, thank you for your nice study regarding portosystemic shunts in dogs. It would be nice, if you could make some amendments in your manuscript.
Line 125 (Strickland 2018) please correct to reference number 16
Line 133 : polypropylene suture- company name?
Line 135: double folded ? why not triple folded as published (Mc Alinden et al, Vet Surg 2010)
Were different kinds of cellophane used?
Line 256 (P=0.142) change to (p=..
QOL score quite high, is the assessment specific enough? Even dogs that have been operated on twice and a high QOL?
Please discuss questionnaire, not anonymized
Kind regards
Author Response
Response to Reviewer 1 attached

Reviewer 2 Report
Dear authors, this is a nicely-written prospective paper comparing the outome of 110 dogs undergoing complete polypropylene ligation versus partial film band attenuation of extrahepatic shunts. I believe that the results of the present study will be of interest to surgeons and dog owners. The authors should include in the discussion the rationale behind the use of polypropylene versus silk used for shunt ligation. Perhaps silk may induce an inflammatory reaction resulting in a faster shunt closure. Other minor comments mainly editorial are listed below:
Introduction
L. 67 ........narrowing reported........
L. 100 ........outcome of dogs..........
L.102 ......compare the outcome between......
M&M
L. 130......ETCO2
L. 148 .......follow up bile acids..........
Quality of English is excellent
Author Response
Response to Reviewer 2 attached

Reviewer 3 Report
To begin with, I congratulate you because this is a very good and interesting prospective study, well structured and with clear and practical clinical conclusions. I really only have some minor flaws that in my opinion should be corrected.
- Line 121: the bibliographic citation referring to White and Perry 2018 is not reflected in the References section.
- Line 126: in the same way as the previous comment occurs with the quote referring to Strickland 2018. On the other hand, in the rest of the work the bibliographic citations are arranged in numbers.
- Line 132-134: This sentence is a bit confusing to read, maybe I shouldn't repeat the word "during" twice.
- Line 161: It would not be necessary to say all the words again, just the acronym.
- Line 186: In this case he uses only the acronym IOMP without having written its meaning beforehand.
- Line 200: I think it would be more correct to start the paragraph in another way and not with numbers.
- Try to unify criteria, since on some occasions they write the "p" of significance with lower case and on others with upper case.
- In the Discussion section they rename the meaning of the acronyms and in my opinion it would not be necessary.
Author Response
Response to Reviewer 2 attached
